# Correcting the Bias of Normalizing Flows by Synthetic Outliers for Improving Out-of-Distribution Detection

## Abstract

Out-of-distribution (OOD) detection is critical for ensuring the reliability and robustness of deep learning models in real-world applications. While normalizing flows have demonstrated impressive performance for various task of image OOD detection, recent findings suggest that they still encounter limitations and severe biases when applied to datasets with different statistics. Specifically, it has been observed that normalizing flow models tend to assign higher likelihoods to OOD samples with low complexity, which undermines the effectiveness of likelihood based OOD detection methods. In this paper, we explore the bias related to data complexity linked to normalizing flow models in OOD detection. We propose a novel method for bias correction by incorporating synthetic outliers during training, guiding the model to assign lower likelihoods to OOD samples. Additionally, we introduce a specialized training objective that leverages the softplus function for OOD data, ensuring a smooth and effective training process. Extensive experiments on benchmark and high-dimensional real-world datasets, including both images and texts, confirm that our proposed approach significantly enhances OOD detection accuracy, achieving performance comparable to models trained with a limited number of real outliers. Moreover, our method increases the Lipschitz constant, supporting the hypothesis presented in related literature.

## 1 Introduction

Out-of-distribution (OOD) detection is essential for ensuring the reliability and safety of deep learning models, especially in real-world applications such as autonomous driving, medical diagnosis, and security systems (Chandola et al., 2009; Hendrycks & Gimpel, 2016; Chalapathy & Chawla, 2019; Cao et al., 2020; Yang et al., 2021; Pang et al., 2021; Morteza & Li, 2022). These systems often encounter data outside the training distribution, and the failure to accurately detect such OOD data can lead to severe consequences. Therefore, OOD detection has become a critical area of research, aimed at improving machine learning models' ability to effectively distinguish between in distribution (ID) and OOD data.

Normalizing flows are recognized as a powerful tool in deep learning due to their ability to model complex probability distributions and perform exact likelihood estimation (Dinh et al., 2014; 2016; Kingma & Dhariwal, 2018; Papamakarios et al., 2021). By leveraging invertible transformations, normalizing flows enable flexible, bijective mappings from data space to a latent space, making them effective in extracting features from in-distribution data. The general principle is to train normalizing flows mapping on high level features of ID data by maximizing the likelihood and then use low likelihood as a score function for OOD detection(Nalisnick et al., 2018; Serrà et al., 2020; Kirichenko et al., 2020; Zhang et al., 2021; Ahmadian & Lindsten, 2021; Osada et al., 2024). They have already demonstrated strong performance in anomaly detection, covering time series, industrial imaging, and medical imaging (Yu et al., 2021; Rudolph et al., 2021; 2022; Dai & Chen, 2022; Gudovskiy et al., 2022; Zhao et al., 2023).

However, recent literature report that there exhibits likelihood bias with normalizing flows based model for OOD detection. Ren et al. (2019) observed that likelihood scores are significantly influenced by population-level background statistics and proposed using a likelihood ratio test for OOD

detection. Nalisnick et al. (2019) introduced the concept of the typical set to determine whether an input is OOD. Additionally, Serrà et al. (2020) demonstrated that simpler images tend to yield higher likelihoods, irrespective of whether the data is OOD, and proposed a new OOD detection score that accounts for input complexity to mitigate this effect. Osada et al. (2024) presented a hypothesis offering a theoretical explanation for the likelihood bias due to image complexity and the reason for the failure of OOD detection.

While normalizing flows are unsupervised models that learn exclusively from in-distribution data, recent studies have explored the incorporation of outlier data to regularize the model and improve its performance in OOD detection. For example, Outlier Exposure (OE), proposed by Hendrycks et al. (2018), introduced auxiliary outliers during training to improve the model's ability to differentiate between ID and OOD data. Expanding on this, Virtual Outlier Synthesis (VOS), introduced by Du et al. (2022), generated synthetic outliers through empirical Gaussian sampling in an unsupervised setting, thereby improving the model's generalization to unseen data. Moreover, Wang et al. (2023) constructed an OOD distribution set that encompasses all distributions within a Wasserstein ball centered on the auxiliary OOD distribution, while Zheng et al. (2023) proposed using generated data to design an auxiliary task for improved OOD detection. Additionally, SANFlow (Kim et al., 2023) incorporated synthetic outliers and trained normalizing flows across multiple distributions instead of a single distribution to enhance anomaly detection performance.

Based on the observations that bias occurs in normalizing flows based OOD detection methods, where performance is robust when the ID data is less complex than the OOD data but declines when the ID data is more complex, we aim to counteract this bias and enhance models OOD detection ability for a wide range of applications. Specially, we propose incorporating synthetic outliers into the training process to reduce the likelihood bias. For both image and text data, we propose a simple but effective synthetic method for outliers generation from ID data. An adverse likelihood objective, which simultaneously maximizes the likelihood of ID samples while minimizing that of OOD samples is proposed for training normalizing flows. By incorporating the softplus function, this objective ensures numerical stability and prevents gradient explosion, allowing for smoother optimization. Furthermore, our approach is also shown to ensure that the trained normalizing flow models satisfy broader Lipschitz continuity conditions, a crucial assumption in prior work, which validates the model's ability to prevent the assignment of higher likelihoods to low-complexity OOD inputs. Comprehensive experiments on widely-used benchmark datasets and high-dimensional real-world datasets, including texts and images, demonstrating that our method significantly improves OOD detection performance. Notably, our approach with synthetic outliers are comparable to the use of limited real labelled outlier data, which demonstrates the effectiveness of the proposed bias correction method for normalizing flows using synthetic outlier data.

The remainder of the paper is organized as follows: Section 2 describes our methodology, starting with the hypothesis and motivation, followed by the generation of synthetic outliers for both images and texts. We then discuss the learning objective and OOD scoring method. Section 3 presents our experimental setup and results, covering benchmark image datasets, high-dimensional image datasets, and text datasets. Finally, Section 4 concludes the paper with a summary of key findings.

## 2 METHODOLOGY

### 2.1 HYPOTHESIS AND MOTIVATION

Our methodology is motivated by a hypothesis from Osada et al. (2024), which establishes a relationship between the complexity of an input sample $\mathbf{x}$ in the data space $\mathcal{X}$ and its latent representation $\mathbf{z}$ in the latent space $\mathcal{Z}$. We first define the concepts of image and text complexity. Notably, the notion of image complexity is based on the definition provided in Serrà et al. (2020); Osada et al. (2024).

**Definition 1** (Image Complexity). *An image $\mathbf{x} \in \{0, 1, \dots, 255\}^d$ is represented as a vector of $d$ pixel values. Let $L(\mathbf{x})$ denote the length in bits of the compressed bit string obtained by applying a lossless compression algorithm (denoted as comp) to $\mathbf{x}$. The image complexity is defined as:*

$$C(\mathbf{x}) = \frac{1}{d}L(\mathbf{x}).$$

*A higher value of $C(\mathbf{x})$ indicates a more complex image $\mathbf{x}$, while a lower value indicates less complexity.*

In the case of text complexity, longer sentences are typically associated with higher complexity. Therefore, we assess the complexity of the entire dataset using a lossless compression algorithm, such as gzip. The text complexity of the dataset $X$ is also defined as $C(X) = \frac{1}{d}L(X)$, where $L(X)$ denotes the length in bits of the compressed data obtained by applying a lossless compression algorithm to the dataset $X$, and $d$ is the number of text in the dataset $X$.

Based on the definition of the complexity of inputs, Hypothesis 1 can be formulated as follows:

**Hypothesis 1** (Osada et al. (2024))**.** *Let $f : \mathcal{X} \to \mathcal{Z}$ be an invertible function that is locally $L_{\mathcal{A}}$-Lipschitz on a subset $\mathcal{A} \subseteq \mathcal{Z}$. For every $\mathbf{z}' \in \mathcal{A}$, define $\mathbf{x}' = f^{-1}(\mathbf{z}')$. Fix a point $\mathbf{z} \in \mathcal{A}$ and choose a constant $\epsilon > 0$ such that the open ball $\mathcal{B}_{\mathbf{z}}^{\epsilon} = \{\mathbf{z}' \in \mathcal{A} \mid \|\mathbf{z}' - \mathbf{z}\| < \epsilon\}$ is contained within $\mathcal{A}$. Let $C(\mathbf{x})$ denote the image complexity of $\mathbf{x} = f^{-1}(\mathbf{z})$, and let $C_1$ be a positive constant. Then, the following inequality holds:*

$$\frac{\epsilon^2}{L_{\mathcal{A}}^2} \left(1 - \mathbb{P}\left(\mathcal{B}_{\mathbf{z}}^{\epsilon}\right)\right) \leq C_1 \exp\left(C(\mathbf{x})\right). \tag{1}$$

Building on Hypothesis 1, we note that a less complex input $\mathbf{x}$ leads to a latent representation $\mathbf{z}$ concentrated in high-density regions of the latent space $\mathcal{Z}$, resulting in a higher likelihood $\log p(\mathbf{z})$. Therefore, normalizing flows tend to perform well in OOD detection when the ID data is of lower complexity and the OOD data is of higher complexity. However, when the ID data is highly complex and the OOD data is simpler—the performance of normalizing flows deteriorates. Table 1 illustrates this phenomenon, showing that the detection performance improves when the complexity of the ID data is lower than that of the OOD data.

Table 1: AUROC scores for OOD detection with varying ID and OOD datasets. The first column represents the ID dataset used for training, while the first row indicates the OOD dataset used for evaluation. The image complexity of the datasets is ordered as SVHN < CIFAR10 < iSUN. The results show that when the complexity of the ID data is lower than the OOD data, the detection performance improves.

|  | CIFAR10 | SVHN | iSUN |
|---|---|---|---|
| **CIFAR10** | – | 44.3 | 73.2 |
| **SVHN** | 94.8 | – | 97.7 |
| **iSUN** | 55.5 | 57.8 | – |

To address this limitation, we propose incorporating synthetic outliers during training, specifically designed to correct the bias toward simpler OOD inputs. By introducing these synthetic outliers, we aim to reduce the likelihood assigned to low-complexity OOD samples, thereby improving the model's ability to differentiate between ID and OOD data.

Moreover, as highlighted in Remark 2 of Osada et al. (2024), the local Lipschitz constant $L_{\mathcal{A}}$ increases as the complexity of the input decreases. This implies that the mapping function $f$ becomes more sensitive to variations in the latent space when processing less complex inputs. To the best of our knowledge, this relationship is model-dependent and has not been explicitly explored in the context of OOD detection in the existing literature.

From this perspective, we hypothesize that the introduction of synthetic outliers for training has an influence on the local Lipschitz constant $L_{\mathcal{A}}$, increasing its value and leading to more dispersed latent representations for low-complexity OOD images. To validate the effectiveness of our method, we will empirically measure changes in $L_{\mathcal{A}}$ by assessing the gradient norms of the model. If our method leads to both improved experimental results and an increase in $L_{\mathcal{A}}$, it would confirm the utility of introducing synthetic outliers to correct the bias in normalizing flows.

## 2.2 SYNTHETIC OUTLIERS

In order to regularize the normalizing flow model, we introduce synthetic OOD samples. These synthetic OOD samples are crafted based on the empirical observation that normalizing flow models tend to assign higher likelihoods to data with lower complexity. Our approach defines synthetic outliers as low-complexity samples that are distinct from the ID data while still retaining certain structural elements. In the following, we present the method of synthesize outliers of low complexity

for both image and text data. Incorporating these outliers during training encourages the model to assign lower likelihoods to OOD data and enhances its ability to distinguish between ID and OOD samples. Let $\mathcal{X}$ denote the ID domain and $\mathcal{Z}$ the corresponding latent space.

### 2.2.1 IMAGE OUTLIERS SYNTHESIS

For a given ID sample $\mathbf{x} \in \mathcal{X}$, we generate a synthetic outlier $\mathbf{x}' \in \mathcal{X}$. For images, we initially generate an augmented outlier $\mathbf{x}^{\mathrm{a}} \in \mathcal{X}$ by randomly employing one of three techniques: CutPaste(Li et al., 2021), CutMix(Yun et al., 2019), or MixUp(Zhang, 2017). This step enriches the semantic diversity and broadens the range of the outliers. Then we synthesize an outlier by applying a Gaussian blur to the augmented outlier, which effectively reduces image complexity and introduces a controlled degree of randomness. The Gaussian blur is a widely used technique in image processing that smooths an image by replacing each pixel with the average of the pixels within a certain radius. The augmentation and Gaussian blur processes are applied independently and randomly. The mathematical formulation of the convolution operation used for generating the synthetic outlier in a multi-channel image is defined as follows:

$$\mathbf{x}'_c(u, v) = (\mathbf{x}^{\mathrm{a}}_c * \mathbf{g})(u, v) = \sum_{i=-k}^{k} \sum_{j=-k}^{k} \mathbf{x}^{\mathrm{a}}_c(u - i, v - j)\, \mathbf{g}(i, j), \quad \forall c \in \{1, 2, \ldots, C\}$$

In this formulation, $C$ is the number of channels in the image (e.g., $C = 3$ for RGB images). The variable $\mathbf{x}^{\mathrm{a}}_c$ denotes the $c$-th channel of the augmented image $\mathbf{x}^{\mathrm{a}}$, $\mathbf{g}$ is Gaussian blur kernel which has dimensions of $(2k + 1) \times (2k + 1)$, and $\mathbf{x}'_c(u, v)$ represents the output value at position $(u, v)$ in the $c$-th channel after applying convolution operation.

By convolving the augmented image $\mathbf{x}^{\mathrm{a}}$ with the Gaussian kernel, we reduce its high-frequency components, thereby generating a synthetic version that retains the global structure but reduces the image complexity. It is crucial to carefully tune the blur degree to balance two competing objectives: (1) ensuring that the synthetic outliers are sufficiently different from the ID samples to be effective for training, and (2) avoiding excessive distortion that would render the outliers unrecognizable and irrelevant for model training. Achieving this balance is critical for the model to robustly distinguish between authentic data and artificially degraded samples, thereby enhancing its ability to detect outliers.

### 2.2.2 TEXT OUTLIERS SYNTHESIS

To generate synthetic text outliers, we employ a two-step process focusing on both sentence length and vocabulary complexity. First, we tokenize the input text into individual sentences and filter out longer sentences by applying a maximum length constraint. Sentences exceeding this length are removed, enhancing the overall readability and reducing complexity.

Next, we simplify the vocabulary by substituting complex words with simpler synonyms. For each word in the remaining sentences, we identify its part of speech and attempt to find a synonym using WordNet (Miller, 1994). We replace the word with the first synonym that differs from the original word. If no suitable synonym is found, the original word is retained. This process preserves the semantic content of the text while simplifying its vocabulary.

The simplified sentences are then concatenated to form a new text, creating a synthetic outlier that is less complex than the original text. This approach effectively generates synthetic text outliers that can be utilized in our OOD detection framework, helping the model distinguish between ID and OOD samples.

### 2.3 LEARNING OBJECTIVE

Our learning objective of the normalizing flow model $f : \mathcal{X} \to \mathcal{Z}$ is designed to maximize the likelihood of ID samples while simultaneously minimizing the likelihood assigned to synthetic OOD samples. Here, $\mathcal{X}$ represents the input space (i.e., the observed data space), and $\mathcal{Z}$ represents the latent space, where the distribution is typically modeled as a standard normal distribution, $p_{\mathcal{Z}}(\mathbf{z}) = \mathcal{N}(\mathbf{0}, \mathbf{I})$. Our adverse objective is formulated as a combination of the maximum likelihood estimation for the ID samples and a softplus-based penalty for the OOD samples.

For ID samples $\mathbf{x} \in \mathcal{X}$, the normalizing flow model seeks to maximize the likelihood of the observed data under the model. The likelihood is computed by transforming the data into the latent space and applying the change-of-variables formula.

The mathematical formulation of the objective function is as follows:

$$\mathcal{L}_{\text{ID}} = -\log p_{\mathcal{X}}(\mathbf{x}) = -\log p_{\mathcal{Z}}(\mathbf{z}) - \log \left| \det \frac{\partial \mathbf{z}}{\partial \mathbf{x}} \right|, \qquad (2)$$

where $p_{\mathcal{X}}(\mathbf{x})$ is the probability density of the ID sample $\mathbf{x} \in \mathcal{X}$ under the model, $p_{\mathcal{Z}}(\mathbf{z})$ is the probability density in the latent space, and the Jacobian determinant $\left| \det \frac{\partial \mathbf{z}}{\partial \mathbf{x}} \right|$ captures the change of variables between the input space and the latent space.

Conversely, for OOD sample $\mathbf{x}' \in \mathcal{X}$, the model aims to assign low likelihoods $p_{\mathcal{X}}(\mathbf{x}')$, encouraging the separation between ID and OOD data. However, as $p_{\mathcal{X}}(\mathbf{x}') \to 0$, the logarithm $\log p_{\mathcal{X}}(\mathbf{x}') \to -\infty$, which can cause numerical instability during optimization. To mitigate this issue, we introduce a softplus minimization term that smoothly penalizes low probabilities assigned to these synthetic outliers while maintaining numerical stability. Applying softplus function to $\log p_{\mathcal{X}}(\mathbf{x}')$, we have:

$$\text{Softplus}(\log p_{\mathcal{X}}(\mathbf{x}')) = \log(1 + \exp(\log p_{\mathcal{X}}(\mathbf{x}'))) = \log(1 + p_{\mathcal{X}}(\mathbf{x}')).$$

As $\log p_{\mathcal{X}}(\mathbf{x}') \to -\infty$ (i.e., $p(\mathbf{x}') \to 0$), the Softplus function approaches $\log(1+0) = 0$, effectively penalizing OOD samples with very low likelihoods without causing numerical instability. Since $\log p_{\mathcal{X}}(\mathbf{x}')$ is a function of the high-dimensional input $\mathbf{x}'$, we compute the full gradient with respect to $\mathbf{x}'$ using the chain rule:

$$\nabla_{\mathbf{x}'} \text{Softplus}(\log p_{\mathcal{X}}(\mathbf{x}')) = \frac{p_{\mathcal{X}}(\mathbf{x}')}{1 + p_{\mathcal{X}}(\mathbf{x}')} \cdot \nabla_{\mathbf{x}'} \log p_{\mathcal{X}}(\mathbf{x}').$$

Here, $\nabla_{\mathbf{x}'} \log p_{\mathcal{X}}(\mathbf{x}')$ represents the gradient of $\log p_{\mathcal{X}}(\mathbf{x}')$ with respect to $\mathbf{x}'$, which is a vector whose components depend on the specific form of $p_{\mathcal{X}}(\mathbf{x}')$. The scalar factor $\frac{p_{\mathcal{X}}(\mathbf{x}')}{1 + p_{\mathcal{X}}(\mathbf{x}')}$ serves to modulate the gradient $\nabla_{\mathbf{x}'} \log p_{\mathcal{X}}(\mathbf{x}')$. For small values of $p_{\mathcal{X}}(\mathbf{x}')$ of outliers, the factor approaches 0 and this property facilitates stable and effective optimization during training by preventing extreme gradient values that could destabilize the learning process, especially in high-dimensional settings.

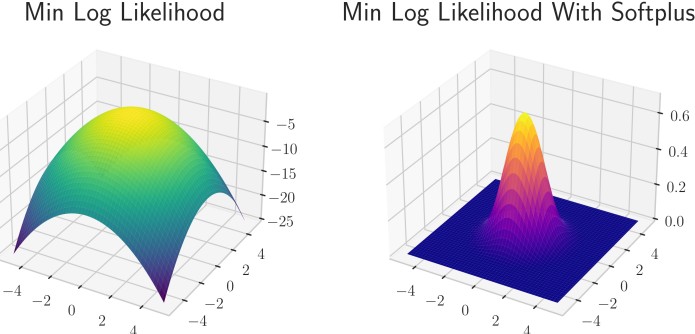

Figure 1: Comparison of log-likelihood loss for a standard normal distribution before and after the Softplus transformation. The left plot illustrates the log-likelihood loss for a normal distribution, where the loss becomes negatively unbounded at the tails, leading to potential numerical instability during training. In contrast, the right plot shows the loss after applying the Softplus function, which produces a smooth and bounded loss surface, thereby improving the training stability.

Figure 1 illustrates the comparison between with and without the softplus function on the logarithm likelihood of standard normal distribution, demonstrating the smooth and stable behavior it induces. The corresponding objective is formulated as:

$$\mathcal{L}_{\text{OOD}} = \text{Softplus}(\log p_{\mathcal{X}}(\mathbf{x}')) = \log\left(1 + p_{\mathcal{X}}(\mathbf{x}')\right) = \log\left(1 + p_{\mathcal{Z}}(\mathbf{z}') \left| \det \left( \frac{\partial \mathbf{z}'}{\partial \mathbf{x}'} \right) \right| \right), \qquad (3)$$

where $\mathbf{x}' \in \mathcal{X}$ is a synthetic outlier generated from the ID sample, and $p_{\mathcal{X}}(\mathbf{x}')$ is the model-assigned probability of the OOD sample. The softplus function ensures that the model learns to reduce the likelihood of these outliers without introducing numerical instability. We notice that Schmier et al. (2022) designed the OOD likelihood loss to train the normalizing flows for contrastive data. To avoid unbounded OOD loss, they manually set a threshold to clamp the loss. While in our work, the specialized training objective for OOD samples ensures smooth and robust training without choosing a manual threshold.

The total learning objective combines these two components:

$$\mathcal{L}_{\text{total}} = \mathcal{L}_{\text{ID}} + \mathcal{L}_{\text{OOD}}.$$

Instead of utilizing weights to balance the two loss function, we choose to adjust the random probability of generating outlier points during the data loading phase. This strategy allows us to directly control the proportion of ID and OOD samples in the training data. By fine-tuning the outlier synthesis probability empirically, we achieve an optimal balance between maximizing the likelihood of ID samples and minimizing the likelihood of OOD samples.

For illustration, Figure 2 demonstrates the performance of normalizing flows trained with and without OOD loss. In the moon dataset, the model incorporating OOD samples assigns lower likelihoods to regions outside the distribution, whereas the model trained exclusively on ID samples assigns high likelihoods to the area linking the two moons. A similar phenomenon is observed in the circle dataset: the model incorporating OOD samples correctly assigns lower likelihoods to OOD regions, while the model trained only on ID samples assigns high likelihoods to the central OOD regions. The details of the experiments are described in Appendix A.

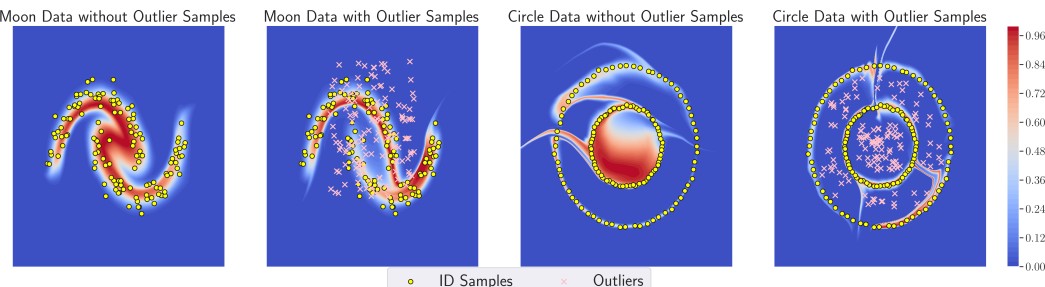

Figure 2: Learned likelihood of normalizing flows trained without and with OOD samples. Redder regions correspond to higher likelihood values, with the models trained with OOD samples more effectively concentrating high likelihoods around the ID data.

## 2.4 OOD SCORING

To determine whether a given sample is OOD or ID, we employ two approaches for computing OOD scores:

1. **Likelihood-based Scoring**: This method leverages the negative log-likelihood derived from a trained normalizing flow model to quantify how well a sample $\mathbf{x}$ fits the learned data distribution. The negative log-likelihood, $\mathcal{S}_{\text{nll}}$, serves as a measure for anomaly detection, with lower values indicating better conformity to the distribution:

$$\mathcal{S}_{\text{nll}}(\mathbf{x}) = -\log p_{\mathcal{Z}}(\mathbf{z}), \tag{4}$$

where $\mathbf{z}$ is the latent representation of $\mathbf{x}$ in the learned feature space and $p_{\mathcal{Z}}(\mathbf{z})$ is the density under the model.

2. **Complexity-adjusted Scoring**: To account for the inherent complexity of the input data, we extend the likelihood-based scoring by incorporating an image complexity term, as proposed by Serrà et al. (2020). The complexity-adjusted score, $\mathcal{S}_{\text{comp}}$, is defined as:

$$\mathcal{S}_{\text{comp}}(\mathbf{x}) = \mathcal{S}_{\text{nll}}(\mathbf{x}) - C(\mathbf{x}), \tag{5}$$

where $C(\mathbf{x})$ denotes the complexity of the image $\mathbf{x}$. This adjustment ensures that the score better reflects images with varying levels of complexity.

The decision function $h : \mathcal{X} \rightarrow \{0, 1\}$ is introduced to classify samples based on their OOD score. Specifically, $h$ maps a sample $\mathbf{x}$ to either 1 or 0, depending on whether the score exceeds a threshold $\gamma$. The decision rule is defined as:

$$h(\mathbf{x}) = \begin{cases} 1 & \text{if } \mathcal{S}(\mathbf{x}) \geq \gamma, \\ 0 & \text{if } \mathcal{S}(\mathbf{x}) < \gamma, \end{cases} \tag{6}$$

where $h(\mathbf{x}) = 1$ indicates that the sample is classified as OOD, and $h(\mathbf{x}) = 0$ indicates that the sample is classified as ID. The threshold $\gamma$ can be selected for optimal F1-score or to ensure that a high proportion of ID samples (e.g., 95%) are correctly classified. This scoring framework provides a flexible and robust mechanism for OOD detection across various datasets and applications, allowing for effective detection in complex real-world scenarios.

## 3 EXPERIMENTS

### 3.1 BENCHMARK IMAGE DATASETS

In our experiments, we aim to evaluate the effectiveness of training normalizing flow models using synthetic outliers and the adverse likelihood loss function. For images, we employ JPEG2000 compression to calculate the image complexity of the datasets. We utilize CIFAR-10, CIFAR-100(Krizhevsky et al., 2009), SVHN(Netzer et al., 2011), LSUN(Yu et al., 2015), iSUN (Xu et al., 2015) and CelebA(Liu et al., 2015) for our experiments, and the details of the complexities can be found in Appendix B. We employ the CIFAR-10, CIFAR-100, and iSUN datasets as ID datasets due to their relatively high complexity. For each dataset, we calculate two commonly used metrics in OOD detection: AUROC and FPR95.

The normalizing flow model is implemented using the FrEIA library (Ardizzone et al., 2018-2022). Each coupling layer's subnet is composed of two $3 \times 3$ convolutional layers with ReLU activations. The model consists of 8 coupling layers, utilizing the `AllInOneBlock` module from the FrEIA library (Ardizzone et al., 2018-2022). The model is trained for 500 epochs on the CIFAR-10 and CIFAR-100 datasets, with synthetic outliers generated at a probability of 0.5. The hyperparameter $\alpha$ for CutMix and MixUp is set to be 1.0. All input images are resized to $32 \times 32$. We employ the Adam optimizer (Kingma, 2014) with a learning rate of $1 \times 10^{-3}$, a weight decay of $1 \times 10^{-5}$, and a batch size of 128. For evaluation, we randomly select 1,000 ID and 1,000 OOD samples to ensure a balanced evaluation dataset. The radius setting of the Gaussian filter is 1. The model's performance is evaluated on the OOD datasets using both likelihood-based scoring and complexity-adjusted scoring methods, aiming to quantify its efficacy in detecting OOD samples under various conditions.

The results are presented in Table 2 and Table 3, where we evaluate the effectiveness of the synthetic outliers and dual likelihood objective. These methods include maximum likelihood estimation trained solely on ID data (MLE), dual likelihood training with a few real outliers (RO) (specifically comprising 10% of the ID data samples), and dual likelihood training using Gaussian blur, the CCM augmentaion (which combines CutPaste, CutMix, and MixUp), and a combination of both Gaussian blur and CCM. The variant methods referred to as 'Methods + Complexity' utilize a complexity-adjusted score as the metric for out-of-distribution (OOD) detection. The results clearly demonstrate that our proposed method significantly outperforms methods trained solely on ID data using MLE. Moreover, in some OOD datasets, synthetic outliers matches or even exceeds the performance of methods that utilize real outliers, achieving the lowest FPR95 and the highest AUROC scores. While the addition of a complexity-based OOD score performs well in certain scenarios, it becomes misleading on OOD datasets with higher complexity due to inherent bias in the score itself ($-C(\mathbf{x})$ will be lower). This can be observed in Table 2 when using CIFAR-10 or CIFAR-100 as ID sets and iSUN as OOD testing set, where the AUROC is exceedingly low. Therefore, the application of complexity-adjusted score should be context-dependent. Overall, the strategy of incorporating synthetic outliers typically reflects the previously described improvements in both robustness and generalization capabilities.

Figure 3 illustrates the relationship between image complexity $C(\mathbf{x})$ and latent space likelihood $p_{\mathcal{Z}}(\mathbf{z})$. In this case, the CIFAR dataset represents ID data, while SVHN and iSUN are the corresponding OOD samples. The pink line represents the linear regression of these points. All data points are obtained from the test sets of the respective datasets. Without outlier training, a clear

Table 2: Comparison of various methods on CIFAR as the ID dataset and SVHN, LSUN, iSUN, and CelebA as OOD datasets.

| Method | SVHN | | LSUN | | iSUN | | CelebA | |
|---|---|---|---|---|---|---|---|---|
| | FPR95↓ | AUROC↑ | FPR95↓ | AUROC↑ | FPR95↓ | AUROC↑ | FPR95↓ | AUROC↑ |
| **CIFAR-10** | | | | | | | | |
| MLE | 95.4 | 44.3 | 41.1 | 86.6 | 74.5 | 73.2 | 58.1 | 76.4 |
| RO | 87.2 | 68.7 | 46.2 | 81.0 | 31.9 | 95.2 | 58.1 | 76.3 |
| Gaussian | 93.0 | 74.6 | 33.8 | 85.0 | 39.1 | 89.9 | **57.7** | **79.7** |
| CCM | 96.1 | 40.1 | 65.4 | 81.4 | **0.0** | 99.3 | 56.7 | 78.8 |
| CCM+Gaussian | **33.4** | **83.2** | **13.9** | **96.0** | 0.2 | **99.4** | 60.7 | 76.1 |
| MLE+Complexity | 38.5 | 89.6 | 62.9 | 86.0 | 98.5 | 23.1 | 72.4 | 66.7 |
| RO+Complexity | 34.9 | 90.5 | 61.1 | 86.9 | 91.8 | 40.0 | 72.4 | 66.6 |
| Gaussian+Complexity | **33.8** | **90.8** | 58.0 | 87.2 | 96.6 | 28.5 | **71.2** | **71.0** |
| CCM+Complexity | 41.0 | 88.8 | **55.2** | **88.4** | **87.0** | **47.1** | 75.4 | 64.5 |
| CCM+Gaussian+Complexity | 36.3 | 89.9 | 59.7 | 86.7 | 89.1 | 45.4 | 77.8 | 62.8 |
| **CIFAR-100** | | | | | | | | |
| MLE | 98.8 | 32.6 | 59.9 | 67.5 | 79.2 | 73.5 | 76.4 | 55.4 |
| RO | 98.4 | 41.7 | 52.7 | 71.7 | 50.1 | 91.2 | 71.4 | 61.9 |
| Gaussian | **57.2** | **84.3** | 41.2 | 80.6 | 54.5 | 85.4 | **65.4** | **71.1** |
| CCM | 97.8 | 55.0 | 76.8 | 76.4 | 0.4 | 98.9 | 70.4 | 67.0 |
| CCM+Gaussian | 77.9 | 75.9 | **38.7** | **89.1** | **0.3** | **99.1** | 68.3 | 65.1 |
| MLE+Complexity | 45.5 | 86.8 | 65.4 | 83.5 | 98.6 | 20.8 | 76.9 | 59.2 |
| RO+Complexity | 47.3 | 86.2 | 63.5 | 84.1 | 92.7 | 33.4 | 77.0 | 59.7 |
| Gaussian+Complexity | **37.4** | **89.4** | 65.5 | 83.9 | 98.5 | 23.2 | 77.7 | **61.0** |
| CCM+Complexity | 43.3 | 87.3 | 62.9 | 85.2 | **84.1** | 45.6 | **75.9** | 60.0 |
| CCM+Gaussian+Complexity | 39.2 | 88.3 | **62.4** | **85.4** | 85.3 | **45.9** | 78.1 | 59.1 |

bias emerges where lower complexity images tend to receive higher likelihoods. However, after incorporating synthetic OOD data during training, this bias is reduced, and the model assigns lower likelihoods to OOD data, improving its ability to differentiate between ID and OOD samples. We further examined the behavior of ID data under our proposed approach by analyzing the predicted likelihoods $p_{\mathcal{Z}}(\mathbf{z})$ for ID test datasets. As shown in Figure 4, we compared models trained with and without synthetic outlier training. The results indicate that the model retains its ability to appropriately represent the in-distribution data.

Table 3: Comparison of various methods on iSUN as the ID dataset and SVHN, LSUN, CelebA, CIFAR-10 and CIFAR-100 as OOD datasets.

| Method | SVHN | | LSUN | | CelebA | | CIFAR-10 | | CIFAR-100 | |
|---|---|---|---|---|---|---|---|---|---|---|
| | FPR95↓ | AUROC↑ | FPR95↓ | AUROC↑ | FPR95↓ | AUROC↑ | FPR95↓ | AUROC↑ | FPR95↓ | AUROC↑ |
| **iSUN** | | | | | | | | | | |
| MLE | 94.6 | 57.8 | 74.6 | 72.8 | 63.2 | 69.8 | 94.6 | 55.5 | 95.9 | 61.7 |
| RO | **69.6** | **70.3** | **0.4** | **99.8** | **29.2** | **91.8** | 96.4 | 53.2 | 94.8 | 64.6 |
| Gaussian | 92.9 | 64.4 | 6.1 | 92.0 | 66.1 | 77.3 | 87.4 | 62.4 | 91.3 | 68.0 |
| CCM | 89.1 | 68.1 | 78.4 | 70.2 | 88.3 | 63.6 | 83.8 | 69.1 | 92.4 | 64.7 |
| CCM+Gaussian | 88.0 | 65.1 | 59.6 | 77.3 | 54.9 | 83.6 | **59.4** | **81.6** | 87.1 | **76.9** |
| MLE+Complexity | 9.5 | 97.8 | 19.9 | 96.0 | 26.3 | 90.2 | 54.8 | 81.4 | 58.9 | 82.5 |
| RO+Complexity | **6.0** | **98.6** | 9.0 | 98.3 | **16.6** | **93.8** | 56.3 | 81.1 | 57.9 | 83.0 |
| Gaussian+Complexity | 9.7 | 97.8 | 22.0 | 95.6 | 23.2 | 90.8 | 54.8 | 82.1 | 55.9 | 83.2 |
| CCM+Complexity | 8.2 | 97.9 | 20.2 | 95.9 | 29.1 | 89.8 | **44.6** | **84.3** | 57.3 | 83.1 |
| CCM+Gaussian+Complexity | 8.0 | 98.0 | 18.4 | 96.0 | 25.7 | 90.7 | 49.4 | 83.1 | **48.8** | **85.1** |

In our experiments, we also estimate the Lipschitz constant of the trained normalizing flow model to validate the assumption made in Hypothesis 1 regarding the model's effectiveness. We apply the following method for estimating the Lipschitz constant:

1. **Sampling Input Samples**: Randomly select $1,000$ samples $\mathbf{x}_i$ from the input distribution to represent the input space.

2. **Computing Gradients**: For each sample $\mathbf{x}_i$, we utilize PyTorch automatic differentiation tool to compute the gradient $\nabla f(\mathbf{x}_i)$ of the model's output $f(\mathbf{x}_i)$ with respect to the input.

3. **Calculating Norms**: Compute the $L^2$ norm $\|\nabla f(\mathbf{x}_i)\|$ of each gradient.

4. **Estimating the Lipschitz Constant**: Approximate the Lipschitz constant $L_{\mathcal{A}}$ by taking the maximum gradient norm: $L_{\mathcal{A}} = \max_i \|\nabla f(\mathbf{x}_i)\|$.

The results, summarized in Table 4, show that training with synthetic outliers significantly increases the Lipschitz constant. This supports the hypothesis that synthetic outliers enhance the local Lip-

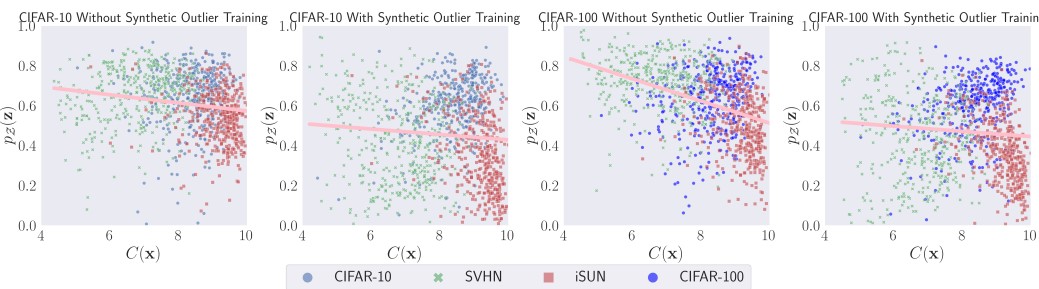

Figure 3: Relationship between complexities $C(\mathbf{x})$ and $p_{\mathcal{Z}}(\mathbf{z})$. Incorporating synthetic outlier training corrects the bias, improving the separation between ID and OOD data.

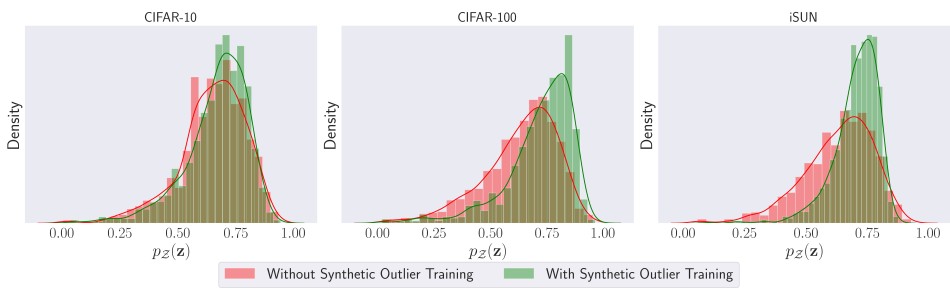

Figure 4: Density estimation plots of $p_{\mathcal{Z}}(\mathbf{z})$ for ID data with and without synthetic outlier training. Synthetic outlier training does not lower the likelihoods of ID data.

schitz constant, improving model stability and performance. To our knowledge, this perspective has not been previously explored. Our study highlights the importance of controlling the Lipschitz constant, thereby addressing a gap in the literature on model robustness in OOD detection for normalizing flows.

Table 4: Significant increases in Lipschitz constants are observed for normalizing flow models trained with synthetic outliers compared to those without.

| $L_{\mathcal{A}}$ | CIFAR10 | CIFAR100 | iSUN |
|---|---|---|---|
| With Synthetic Outlier Training | **7518.26** | **4891.17** | **1700.93** |
| Without Synthetic Outlier Training | 1075.15 | 769.03 | 561.10 |

## 3.2 HIGH-DIMENSIONAL IMAGE DATASETS

We further validate our method using high-dimensional datasets to ensure its applicability in diverse scenarios. The **chest X-ray dataset** (Kermany et al., 2018) was selected due to the inherent difference in visual complexity between pneumonia-infected and normal images—pneumonia images tend to be blurrier and of lower complexity. Additionally, the **RealBlur dataset** (Rim et al., 2020) was used to evaluate the model under realistic blur scenarios. Furthermore, the **KonIQ-10k dataset** (Hosu et al., 2020), which provides image quality assessment scores, allowed us to incorporate image complexity into our evaluation. The complexities and the splitting of ID/OOD data are illustrated in Appendix B. All images are resized to $256 \times 256$. This diverse dataset selection ensured a comprehensive evaluation across image quality and real-world complexity. To evaluate the performance of our approach, we conducted comparisons with two flow-based methods: CS-Flow (Rudolph et al., 2022) and FastFlow (Yu et al., 2021). Specifically, we applied modifications to synthetic outlier strategies and dual likelihood objective, enabling a direct comparison of original method with our approach. We also conducted experiments on the MVTecAD dataset(Bergmann et al., 2019), with detailed results and analyses provided in Appendix C.

The results in Table 5 demonstrate that our approach improves the performance of existing flow-based methods on these datasets. Specifically, both CS-Flow and FastFlow show increased AUROC scores when combined with synthetic outliers. For instance, on the Chest X-ray dataset, CS-Flow's

AUROC increased from 87.7% to 92.3%, and FastFlow improved from 90.3% to 93.4%. Similarly, RealBlur and KonIQ-10k datasets show notable gains, with FastFlow achieving the most substantial improvements, reflecting the effectiveness of our modifications across diverse real-world scenarios.

Table 5: AUROC comparison of different normalizing flow-based methods on the real-world Chest X-ray, RealBlur and konik10k datasets.

| Method | Chest X-ray | RealBlur | KonIQ-10k |
|---|---|---|---|
| CS-Flow (Rudolph et al., 2022) | 87.7 | 70.1 | 62.8 |
| CS-Flow + Gaussian | 88.4 0.7↑ | 75.1 5.0↑ | 64.3 1.5↑ |
| CS-Flow + CCM | 91.7 4.0↑ | 72.7 2.6↑ | 70.8 8.0↑ |
| CS-Flow + CCM + Gaussian | 92.3 4.6↑ | 79.7 9.6↑ | 70.5 7.7↑ |
| FastFlow (Yu et al., 2021) | 90.3 | 77.6 | 71.2 |
| FastFlow + Gaussian | 93.4 3.1↑ | 84.0 6.4↑ | 77.7 6.5↑ |
| FastFlow + CCM | 88.3 2.0↓ | 80.4 2.8↑ | 76.4 5.2↑ |
| FastFlow + CCM + Gaussian | 93.2 2.9↑ | 81.1 3.5↑ | 76.8 5.6↑ |

## 3.3 TEXT DATASETS

As for the text modality, we employ ALBERT Base v2 (Lan, 2019) as the encoder for the text feature embedding. The maximum length constraint is set at 20, and we use the NLTK library (Bird & Loper, 2004) for synonym replacement. We utilize the IMDb dataset (Maas et al., 2011), which exhibits the highest complexity among the datasets considered, as our in-distribution data for training. The remaining datasets—movie reviews (Pang et al., 2002), AG News (Zhang et al., 2015), SST-2 (Socher et al., 2013), and WikiText-2 (Merity et al., 2016)—serve as OOD data. For our experiments, we use 1,000 IMDb samples for training and an additional 1,000 ID and 1,000 OOD samples for testing. The complexities of each dataset is shown in Appendix B. The model architecture follows the normalizing flow used in the image modality, but replaces the convolutional layers in the subnetwork with fully connected linear layers, each with a dimension of 768. We utilize the likelihood-based scoring method and the performance is assessed using two key metrics: AUROC and AUPR.

Table 6: Comparison of AUROC and AUPR values across text datasets.

| Dataset | Without Synthetic Outliers | | With Synthetic Outliers | |
|---|---|---|---|---|
| | AUROC↑ | AUPR↑ | AUROC↑ | AUPR↑ |
| Movie Reviews | 89.8 | 86.4 | 93.9 4.1↑ | 91.4 5.0↑ |
| AG News | 89.1 | 89.0 | 91.8 2.7↑ | 91.6 2.6↑ |
| SST-2 | 63.1 | 66.9 | 98.2 35.1↑ | 98.8 31.9↑ |
| Wiki | 77.4 | 80.4 | 79.2 1.8↑ | 81.8 1.4↑ |

The results in Table 6 show a clear improvement in model performance when synthetic outliers are introduced. Notably, on the SST-2 dataset, both AUROC and AUPR experience substantial increases of 35.1% and 31.9%, respectively, indicating the model's improved capability to handle challenging, low-complexity data. Smaller but consistent gains are observed across other datasets, such as Movie Reviews and AG News, with AUROC improvements of 4.1% and 2.7%, respectively. These results suggest that the introduction of synthetic outliers strengthens the model's generalization and anomaly detection capabilities across diverse textual datasets.

## 4 CONCLUSION

In this work, we propose a novel method to address the bias in normalizing flow models, which tend to assign higher likelihoods to low-complexity OOD samples, reducing their effectiveness in OOD detection. To correct the bias, we incorporate synthetic outliers during training and introduce a specialized objective using the softplus function, enhancing the model's ability to differentiate between ID and OOD data. Extensive experiments on both image and text datasets confirm that our method significantly boosts OOD detection accuracy, achieving performance comparable to models trained with real outliers, while demonstrating broad applicability across tasks. Our approach also improves the model's Lipschitz constant, aligning with the hypothesis that a higher Lipschitz constant enhances robustness in OOD detection.

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

## A   EXPERIMENTS IN FIGURE 2

The experimental setup for generating Figure 2 can be described as follows. The normalizing flow's subnet architecture consists of three fully connected layers, each utilizing ReLU activation functions, with the hidden dimension 256. The model is built using the FrEIA framework's `AllInOneBlock`, employing an eight-step flow procedure.

For the moon dataset, outliers are introduced through random uniform sampling, specifically at coordinates ranging in $(-0.5, 1.5)$. In contrast, for the circles dataset, outliers inside the ring are generated through random uniform sampling within the range $(-0.3, 0.3)$, while those located outside the ring are sampled between concentric circles with radius ranging from 0.6 to 0.9.

Training the model that includes outliers leverages a dual likelihood loss function, whereas the model trained exclusively on in-distribution data utilizes a maximum likelihood loss. Both models are optimized using the Adam optimizer, with a batch size of 128 and a learning rate of $1 \times 10^{-3}$. We complete the training process in $1,000$ epochs for the moon dataset and $5,000$ epochs for the circles dataset, ensuring thorough convergence.

## B   COMPLEXITY OF DIFFERENT DATASETS

In our experiments with the **chest X-ray dataset** (Kermany et al., 2018), we used normal images as ID data for training and treated pneumonia-infected images as OOD data. For the **RealBlur dataset** (Rim et al., 2020), the clear images were designated as ID data, while blurred images were used as OOD data. Additionally, we divided the **KonIQ-10k dataset** (Hosu et al., 2020) based on the MOS z-score, with images scoring higher than 60 considered as ID data and the rest as OOD data.

We provide the detailed complexity of the datasets used in our experiments. For image datasets, complexity is quantified using the mean and variance of the complexity scores, which are computed based on the compression efficiency of each dataset. For text datasets, we present a single complexity value for each dataset, derived from applying a lossless compression algorithm. This uniform metric offers a straightforward comparison of text complexity. Image complexities are shown in Table 7 and Table 8, and text complexities are shown in Table 9. Figure 5 presents the distribution of specific complexity across various benchmark image datasets.

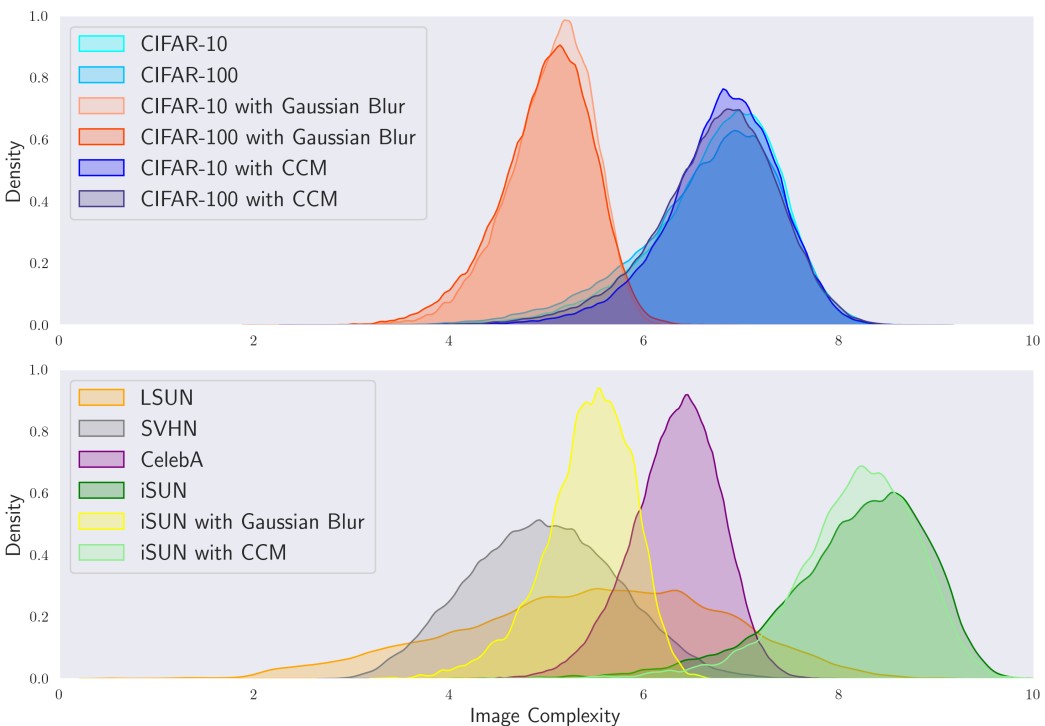

Figure 5: Image complexity distribution across multiple datasets after JPEG2000 compression.

Table 7: Mean and standard deviation of the complexities for image datasets.

| Complexity | SVHN | Blurred CIFAR-100 | Blurred CIFAR-10 | Blurred iSUN | LSUN | CelebA |
|---|---|---|---|---|---|---|
| Mean | 4.97 | 5.00 | 5.04 | 5.43 | 5.44 | 6.33 |
| Standard Deviation | 0.75 | 0.48 | 0.44 | 0.46 | 1.34 | 0.46 |

| Complexity | CIFAR-100 | CIFAR-100 + CCM | CIFAR-10 | CIFAR-10 + CCM | iSUN | iSUN + CCM |
|---|---|---|---|---|---|---|
| Mean | 6.68 | 6.74 | 6.74 | 6.80 | 8.17 | 8.17 |
| Standard Deviation | 0.73 | 0.63 | 0.66 | 0.58 | 0.75 | 0.64 |

Table 8: Mean and standard deviation of the complexities for high-dimensional image datasets.

| Complexity | Chest X-ray | RealBlur | KonIQ-10k |
|---|---|---|---|
| ID | $3.40 \pm 0.21$ | $3.41 \pm 0.55$ | $3.98 \pm 1.09$ |
| OOD | $3.06 \pm 0.29$ | $2.95 \pm 0.55$ | $3.22 \pm 0.90$ |

Table 9: Complexities for different text datasets.

| | Synthesized IMDb | Wiki | AG News | SST-2 | Movie Reviews | IMDb |
|---|---|---|---|---|---|---|
| Complexity | 2.63 | 2.73 | 2.96 | 3.05 | 3.05 | 3.21 |

## C  EXPERIMENTS ON THE MVTECAD DATASET

In this section, we present the performance of our proposed methods on the MVTecAD dataset, which is widely used for anomaly detection in industrial applications. The dataset contains a variety of defect categories, making it a challenging benchmark for evaluating anomaly detection methods.

Our analysis focuses on comparing the performance of FastFlow and CS-Flow, with and without synthetic outlier training (denoted as "+CCM+Gaussian"). We employ Wide-ResNet50-2 as the backbone architecture for FastFlow. The vanilla FastFlow results were obtained through our reproduction of the experiments, while the original results for CS-Flow were directly obtained from the original paper.

Table 10 summarizes the results in terms of AUROC across different categories. The inclusion of synthetic outliers consistently improves performance, particularly in challenging categories such as "screw" and "toothbrush" where substantial gains are observed. For example, FastFlow+CCM+Gaussian achieves an AUROC of 100.0 in categories like "bottle," "carpet," and "grid," outperforming its baseline counterpart. Similarly, CS-Flow+CCM+Gaussian achieves competitive results across all categories, with near-perfect or perfect scores in most cases. The mean scores further validate the effectiveness of synthetic outlier training, as both FastFlow+CCM+Gaussian and CS-Flow+CCM+Gaussian achieve outstanding results, underscoring the robustness and generalizability of our approach.

Table 10: Comparison of AUROC on MVTecAD dataset.

| | FastFlow | FastFlow+CCM+Gaussian | CS-Flow | CS-Flow+CCM+Gaussian |
|---|---|---|---|---|
| bottle | 99.4 | 100.0 0.6↑ | 99.8 | 100.0 0.2↑ |
| cable | 97.6 | 96.5 1.1↓ | 99.1 | 99.1 0.0 |
| capsule | 97.7 | 96.5 1.2↓ | 97.1 | 98.3 1.2↑ |
| carpet | 97.8 | 100.0 2.2↑ | 100.0 | 100.0 0.0 |
| grid | 99.0 | 100.0 1.0↑ | 99.0 | 99.8 0.8↑ |
| hazelnut | 100.0 | 99.3 0.7↓ | 99.6 | 99.7 0.1↑ |
| leather | 100.0 | 100.0 0.0 | 100.0 | 100.0 0.0 |
| metal nut | 99.7 | 100.0 0.3↑ | 99.1 | 98.1 1.0↓ |
| pill | 98.5 | 97.0 1.5↓ | 98.6 | 98.6 0.0 |
| screw | 92.0 | 94.3 2.3↑ | 97.6 | 98.6 1.0↑ |
| tile | 100.0 | 100.0 0.0 | 100.0 | 100.0 0.0 |
| toothbrush | 97.6 | 98.3 0.7↑ | 91.9 | 96.4 4.5↑ |
| transistor | 99.6 | 99.2 0.4↓ | 99.3 | 99.5 0.2↑ |
| wood | 100.0 | 100.0 0.0 | 99.5 | 100.0 0.5↑ |
| zipper | 99.5 | 100.0 0.3↑ | 99.7 | 99.8 0.1↑ |
| mean | 98.6 | 98.7 0.1↑ | 98.7 | 99.2 0.5↑ |

## D    EVALUATION OF BLURRED CIFAR10 AS ID DATASETS

Since we use blurred data as OOD, we conducted experiments where the ID datasets are blurred versions of CIFAR10. We evaluated our method using unblurred original datasets as the OOD scenarios, as well as using other external datasets for OOD. Specifically, we applied Gaussian blur to the CIFAR10 datasets to simulate low-complexity ID data, while the OOD data included both the original unblurred CIFAR10 datasets and other datasets such as SVHN and LSUN. We use `torchvision.transforms.GaussianBlur` in the data preprocessing to simulate blur ID data and the kernel size is set to be 5. While the improvements are not as substantial as those observed in the original dataset from Table 2, the results in Table 11 show a clear enhancement in both AUROC and AUPR metrics with the inclusion of synthetic outliers. This further supports the robustness of our method in handling varying complexities of ID and OOD data.

Table 11: Comparison of AUROC and AUPR values when using blurred CIFAR10 as ID dataset.

| Dataset | Without Synthetic Outliers | | With Synthetic Outliers | |
|---|---|---|---|---|
| | AUROC↑ | AUPR↑ | AUROC↑ | AUPR↑ |
| LSUN | 86.4 | 86.9 | 95.3 8.9↑ | 94.1 7.2↑ |
| SVHN | 44.9 | 46.3 | 47.9 3.0↑ | 47.8 1.5↑ |
| CIFAR10 | 61.0 | 58.1 | 66.7 5.7↑ | 64.2 6.1↑ |

# E    ABLATION STUDY

We conducted an ablation experiment to investigate the effect of varying degrees of blurring on synthetic outliers in images. We use CIFAR-10 and CIFAR-100 datasets as ID data, and SVHN as OOD. The blurring level was controlled by adjusting the `Radius` parameter, with larger values corresponding to higher degrees of blurring. As illustrated in the Figure 6, the results show that as the blurring increases, the AUROC scores initially improve, reaching a peak at moderate blurring levels . However, at extreme levels of blurring, the scores fluctuate and even decrease, indicating that both very low and very high levels of blurring negatively affect the performance of OOD detection methods. This pattern highlights the nuanced impact of blurring on synthetic outliers, with moderate blurring yielding the best detection results.

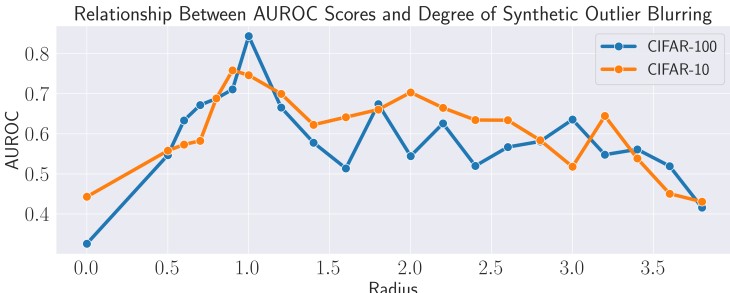

Figure 6: Relationship between AUROC scores and the degree of synthetic outliers blurring, with AUROC peaking at moderate blurring levels and decreasing at extreme blurring levels.

# F    SYNTHETIC SAMPLES

In the section, we provide visual comparisons for several datasets to demonstrate the effectiveness of synthetic outliers. We compare original and synthetic OOD samples in the CIFAR-10 and iSUN datasets, and include both synthetic and real OOD samples for the chest X-ray dataset. For the IMDb dataset, we show how synthetic texts compare with original texts. These examples highlight the utility of our synthetic data across different contexts.

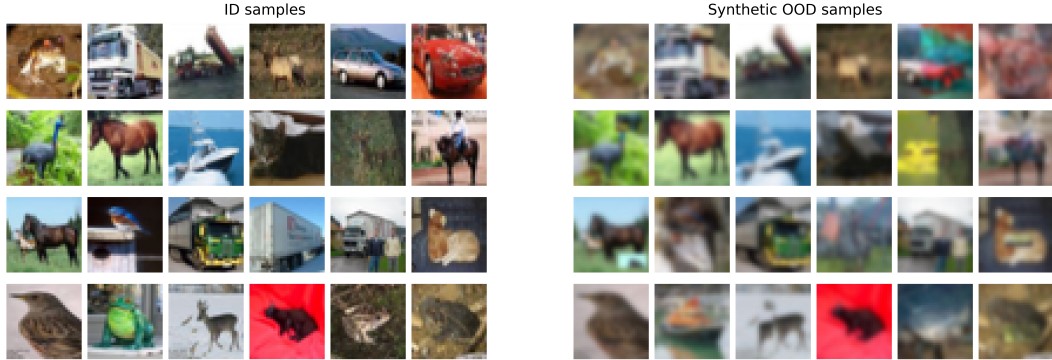

Figure 7: Comparison of original ID and synthetic OOD samples from the CIFAR-10 dataset.

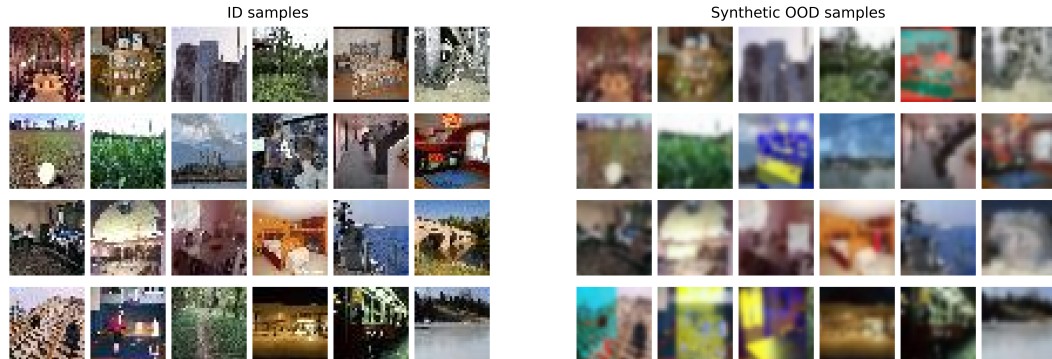

Figure 8: Comparison of original ID and synthetic OOD samples from the iSUN dataset.

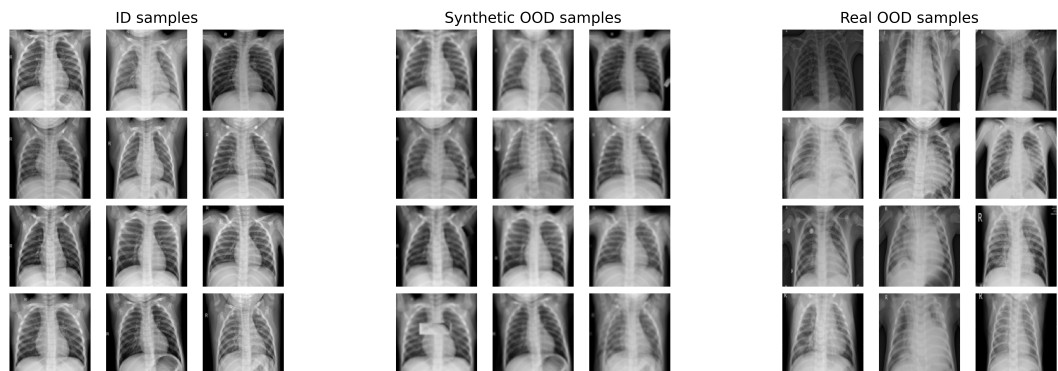

Figure 9: Comparison of original ID, synthetic OOD samples and real OOD samples from chest X-ray dataset.

Table 12: Comparison of original and synthetic texts in IMDb dataset.

| Original ID Text | Synthetic OOD Text |
|---|---|
| My family and I normally do not watch local movies for the simple reason that they are poorly made, they lack the depth, and just not worth our time. The trailer of "Nasaan ka man" caught my attention, my daughter-in-law's and daughter's so we took time out to watch it this afternoon. The movie exceeded our expectations. The cinematography was very good, the story beautiful and the acting awesome. Jericho Rosales was really very good, so's Claudine Barretto. The fact that I despised Diether Ocampo proves he was effective at his role. I have never been this touched, moved and affected by a local movie before. Imagine a cynic like me dabbing my eyes at the end of the movie? Congratulations to Star Cinema!! Way to go, Jericho and Claudine!! | The film exceeded our expectations. The filming was very good, the narrative beautiful, and the acting amazing. Jericho Rosales was really very good, as was Claudine Barretto. The fact that I despised Diether Ocampo proves he was effective in his role. I have never been this touched, moved, and affected by a local film before. Imagine a cynic like me dabbing my eyes at the end of the film? Congratulations to Star Cinema!! Way to go, Jericho and Claudine!! |
| For my humanities quarter project for school, i chose to do human trafficking. After some research on the internet, i found this DVD and ordered it. I just finished watching it and I am still thinking about it. All I can say is "Wow". It is such a compelling story of a 12 year old Vietnamese girl named Holly and an American man named Patric who tries to save her. The ending leaves you breathless, and although itś not a happily-ever-after ending, it is very realistic. It is amazing and I recommend it to anyone! You really connect with Holly and Patric and your heart breaks for her and because of what happens to her. I loved it so much and now I want to know what happens next! | For humanities quarter project school, chose human trafficking. After research internet, found DVD ordered. I finished watching I still thinking. All I say " Wow " . The ending leaves breathless, although 's happily-ever-after ending, realistic. It amazing I recommend anyone! You really connect Holly Patric heart breaks happens. I loved much I want know happens next! |
| Bela Lugosi as creepy insane scientist who uses orchids to woo brides in order to steal life essence for aged wife. The midget in this film is hilarious!! A lot of freaks, plus a lot of padding and no plot makes watching this film a nightmare. I loved how all the pieces fell together in the end in typical Hollywood fashion. The story never gets interesting, and you feel helpless as you watch.¡br /¿¡br /¿Usually I'd score bore flicks like this one low, but the midget added just enough creepiness and entertainment to gain a couple more points. | The dwarf in this movie be screaming!! A batch of freak, plus a batch of embroider and no secret plan make watch this movie a incubus. I love how all the piece drop together in the terminal in typical Hollywood manner. |

