# OpenReview forum: "Correcting the Bias of  Normalizing Flows by Synthetic Outliers for Improving Out-of-Distribution Detection"
_ICLR.cc/2025/Conference — Submitted to ICLR 2025_

### Official Review · Reviewer_SLYb · 2024-10-21

**Soundness:** 3
**Presentation:** 3
**Contribution:** 2
**Rating:** 5
**Confidence:** 5

**Summary:**

- The author finds that normalizing models tend to assign higher likelihood to the input data. Therefore, when the complexity of ID data is lower than that of OOD data, the model will detect OOD samples better. To correct this bias, the author generates synthetic outliers with lower complexity while forcing the model to assign lower likelihood to them.

**Strengths:**

- The paper is well organized and easy to follow.
- The author provides many formulations which help to understand the main pipeline of the manuscript.
- The experiments demonstrate that synthetic outliers improve the detection performance of normalizing flow models.

**Weaknesses:**

- Synthetic outliers is not a novel topic in OOD detection. Similar papers include NPOS (arXiv:2303.02966), VOS (arXiv:2202.01197), SSOD (arXiv:2307.00519), to name a few. Besides, there are also many papers which generate synthetic OOD images to provide auxiliary supervision. Therefore, it seems that this manuscript is not creative enough.
- Lack of comparisons with other OOD synthesis methods.
- The performance is poor compared with SOTA techniques. The OpenOOD benchmark (https://github.com/Jingkang50/OpenOOD) collects main OOD detection methods and demonstrates their FPR95 and AUROC.

**Questions:**

see the weakness.

---

> ### Author Response · Authors · 2024-11-26
>
> Thank you for your detailed and constructive review. Below, we address your concerns:
>
> **Weakness 1**: Synthetic outliers are not a novel topic in OOD detection. Similar papers include NPOS, VOS, SSOD, etc. Therefore, it seems that this manuscript is not creative enough.
>
> **Response**: Synthetic outliers remain a crucial area of study in OOD detection. While this topic has been explored in various contexts, our manuscript focuses on addressing biases related to how these models interact with image complexity. This approach enhances the understanding and application of normalizing flows for detecting outliers, distinguishing our work from existing methodologies and underscoring the ongoing importance and potential for development within this field.
>
> **Weakness 2**: Lack of comparisons with other OOD synthesis methods.
>
> **Response**: In our study, we have expanded our experimental framework to include image space data augmentation techniques such as CutPaste, CutMix, and MixUp. These techniques were selected for their applicability to image space, unlike feature space methods such as NPOS and VOS. We also validate that combining these augmentation techniques with Gaussian blur further improves the results. For example, when using CIFAR-10 as the in-distribution dataset and SVHN, LSUN, and iSUN as the out-of-distribution datasets, the best results are achieved by combining both augmentation techniques and Gaussian blur.
>
> **Weakness 3**: The performance is poor compared with SOTA techniques.
>
> **Response**: Our approach primarily targets correcting biases between normalizing flow models and image complexity. Consequently, our method can be combined with many normlaizing flow based SOTA methods. For real-world datasets like MVTec, we have adopted contemporary data augmentation methods such as CutPaste, and validated our approach using advanced models like FastFlow and CS-Flow. These methods are tailored to enhance performance specifically in practical applications, and our experiments indicate further improvements. The results are shown in Appendix C. We believe that our approach provides important contributions to addressing particular challenges in OOD detection with normalizing flows.

---

> > ### Comment · Reviewer_SLYb · 2024-11-27
> >
> > Your responses have successfully addressed the majority of my concerns. After carefully considering both the original content of the paper and your clarifications in the reply, I have decided to maintain my initial scoring.

---

### Official Review · Reviewer_mLpm · 2024-11-01

**Soundness:** 3
**Presentation:** 3
**Contribution:** 2
**Rating:** 5
**Confidence:** 4

**Summary:**

The paper aims to improve the out-of-distribution detection performance of Normalizing Flows. The paper is motivated by observation from a previous work that generative models, including normalizing flows, assign high likelihood to less complex OOD data, resulting in misclassifying such OOD data as ID data. To address the issue, the proposed method proposes to generate simple OOD data by simply applying Gaussian blur on ID data. Then, the model proposes to minimize the softplus function of likelihood of these synthetic OOD for training stability. As a result, the paper demonstrates non-trivial OOD detection performance improvement upon NF baselines.

**Strengths:**

- The paper is well-written and easy to follow.
- The paper introduces a simple yet effective approach to improve the OOD performance of NF models.
- The proposed idea of generating low-complexity OOD to mitigate previously found drawbacks of NF models is well motivated.

**Weaknesses:**

- The proposed augmentation method is rather too simple that it raises several concerns.
   - There could be several cases in a real-world scenario, where a blurry image is not OOD but ID instead. How to handle such cases?
   - Would this work on more real-world datasets, such as MVTecAD and ShanghaiTech?

- The paper lacks enough discussions on prior works. For example, there is another OOD/anomaly detection paper that extends NF framework design to learn to handle synthetic OOD [A]. Can authors provide discussions and quantitative comparisons against this method (with the same augmentation suggested by the authors or by this paper)?

- In the paper, it says "By fine-tuning the outlier synthesis probability through validation, we achieve an optimal balance between maximizing the likelihood of ID samples and minimizing the likelihood of OOD samples." What is a validation set here? What kind of data is in the validation set? What is its size?

- I'm not really convinced that using synthetic OOD with low complexity is the key. If you generate enough and diverse OOD samples, wouldn't it be helpful even though it's complex? What about applying aug methods that make it more complex (cutmix, mixup, etc)? I believe these kinds of augmentation methods will lead to more complex OOD data but still improve the OOD performance, possibly more than simple Gaussin blurs. The experiments on this could strengthen the authors' claims.


[A] SANFlow: Semantic-Aware Normalizing Flow for Anomaly Detection and Localization, NeurIPS 2023

**Questions:**

I have incorporated questions in the weakness section.

---

> ### Author Response · Authors · 2024-11-26
>
> Thank you for your detailed feedback, which provided valuable insights into the limitations and potential improvements of our work. Below, we address your concerns in detail:
>
> **Weakness 1**: There could be several cases in a real-world scenario where a blurry image is not OOD but ID instead. How to handle such cases?
>
> **Response**: Our method is designed not merely to utilize complexity measures for detecting OOD data but primarily to correct the complexity bias inherent in normalizing flows. This data augmentation method helps to correct the likelihood bias for distinguishing between ID and OOD data more effectively. For OOD data that exhibit similar complexity levels to ID data but differ in semantic content, our approach goes beyond complexity metrics. We conduct experiments with diverse synthetic outliers that simulate a range of complexity and semantic variations.
>
> **Weakness 2**: Would this work on more real-world datasets, such as MVTecAD and ShanghaiTech?
>
> **Response**: We added experiments on MVTec AD datasets to evaluate the generalizability of our method. The results in Appendix C show consistent improvements in anomaly detection, confirming the applicability of our approach to real-world datasets.
>
> **Weakness 3**: The paper lacks enough discussions on prior works. For example, there is another OOD/anomaly detection paper that extends the NF framework design to learn to handle synthetic OOD. Can authors provide discussions and quantitative comparisons against this method (with the same augmentation suggested by the authors or by this paper)?
>
> **Response**: We are enthusiastic about engaging with and discussing significant methodologies such as SANFlow. We have included a discussion of SANFlow in the introduction of our manuscript. We emphasis that our paper primarily introduces methods for synthetic OOD data generation and adjustments to the loss function to enhance training of normalizing flow, rather than developing new network architectures or frameworks. Therefore, we conduct real-world experiments on FastFlow and CS-Flow with our data sugmenmtation method, and demonstrate the effectiveness of synthetic outliers for OOD detection. And this methology is hard to directly compare with methods like SANFlow.
>
> **Weakness 4**: The paper says, "By fine-tuning the outlier synthesis probability through validation, we achieve an optimal balance between maximizing the likelihood of ID samples and minimizing the likelihood of OOD samples." What is a validation set here? What kind of data is in the validation set? What is its size?
>
> **Response**: Thank you for pointing this out. The mention of a validation set was a typographical error in our manuscript. In practice, we fine-tuned the outlier synthesis probability empirically by observing changes in loss and experimental outcomes. We have updated the manuscript to accurately reflect this process.
>
> **Weakness 5**: I'm not really convinced that using synthetic OOD with low complexity is the key. If you generate enough and diverse OOD samples, wouldn't it be helpful even though they're complex? What about applying aug methods that make it more complex (CutMix, MixUp, etc)?
>
> **Response**: We have added experiments with more augmentations like CutPaste, CutMix and MixUp. These methods can be employed simultaneously with Gaussian blur and show improved OOD detection performance, validating the importance of generating diverse synthetic OOD samples to complement the low-complexity approach.
>
> Thank you for your constructive feedback, which helped us broaden the scope of our experiments.

---

> ### Comment · Reviewer_mLpm · 2024-11-27
>
> Thank you for the response.
> Some of my concerns still remain unresolved.
>
> For weakness 1, the response still does not explain the case I mentioned: When ID examples are blurry (which can happen very often), is this method viable? If so, why? If not, the method is not generalizable.
>
> As for the response to weakness 5, why does incorporating CutPaste, CutMix, and MixUp with Gaussian improve the performance? Isn't synthetic OOD with low complexity the key?

---

> > ### Author Response · Authors · 2024-11-27
> >
> > Thank you for the constructive feedback. Below are our responses to the concerns raised.
> >
> > **Response to Weakness 1**:
> > When ID examples are blurry, which corresponds to a scenario with low complexity in the ID data, our method remains effective. As shown in Table 2, when the ID data is of lower complexity, for CIFAR-10 and CIFAR-100 as ID datasets and iSUN as the OOD dataset, performance increases from 73.2 to 89.9 on CIFAR-10 and from 73.5 to 85.3 on CIFAR-100. These results confirm that our method is effective in cases where in-distribution data has low complexity, such as when examples are blurry.
> >
> > **Response to Weakness 5**:
> > Although complexity is indeed a key factor in improving OOD detection for normalizing flows, as the reviewer pointed out, synthetic outliers should exhibit diversity in their characteristics to effectively enhance robustness. To address this, we added experiments (shown in Table 2) where we combined methods such as CutPaste, CutMix, and MixUp with Gaussian blur. While Gaussian blur alone may perform better than these methods on certain datasets (e.g., SVHN and LSUN), the combination of these augmentations leads to significant improvements in performance.
> >
> > For instance, using CIFAR-10 as the in-distribution dataset, the performance on SVHN increases from 74.6 (with Gaussian blur alone) to 83.2 when combined with these augmentations, and on LSUN, performance improves from 85.0 to 96.0. These results demonstrate that combining Gaussian blur with CutPaste, CutMix, and MixUp leverages both low complexity and diversity, ultimately enhancing the model’s ability to detect out-of-distribution samples.

---

> ### Comment · Reviewer_mLpm · 2024-11-27
>
> As for the weakness 1, the response still does not address the question: Why does it work? I thought the paper aims to solve the issue when the "performance ... declines when the ID data is more complex...., by incorporating synthetic outliers into the training process to reduce the likelihood bias" (Line 73-75). ***The most recent comment by authors contradicts the very main claim of this work.*** Furthermore, I wouldn't consider CIFAR-10/100 images to be that blurry. I would have tested by blurring all or a portion of ID training images with Gaussian kernel to simulate blurry ID images. But, I'm not asking for more experiments, as I should not according to the guidelines.
>
> As for the weakness 5, what I'm curious about is why the combination of Gaussian blur and CutPaste, CutMix, and MixUp works better? Don't CutPaste, CutMix, and MixUp introduce more complexities to synthetic OOD data? But, wasn't the claim of the paper that less complex synthetic OOD data is the key?

---

> > ### Author Response · Authors · 2024-11-28
> >
> > Thank you for your response. Below, we provide a clarification in response to your comments.
> >
> > **Weakness 1**: The motivation behind designing synthetic outliers was to reduce the complexity of the data. We chose Gaussian blur because it is an efficient method to reduce the complexity. Our goal is to address the scenario where the ID data is of high complexity and the OOD data is of low complexity. When the reviewer mentioned blurred ID data, we interpret this as corresponding to low-complexity ID and high-complexity OOD scenario, which aligns with the experimental results we provided, where CIFAR is the ID distribution and iSUN is the OOD distribution.
> >
> > Regarding the suggestion of using blurred data as ID, we are not entirely sure whether the OOD should be another dataset or the original unblurred dataset. Therefore, we have conducted experiments under both conditions for the blurred CIFAR10 dataset. The results, which are provided in Tables 11 of Appendix D, show an improvement in performance in both cases.
> >
> > **Weakness 5**: Since CutPaste involves cutting and pasting portions from the same image, and CutMix and MixUp perform mixing or linear combinations within samples from the same ID dataset, the complexity change introduced by these methods is minimal. In the new version of the manuscript, we have updated Figure 5 and Table 7 in Appendix B, which display the complexities of the different datasets. Therefore, this data augmentation does not contradict our original goal of reducing complexity.
> >
> > We hope these clarifications resolve the concerns raised. Thank you again for your  feedback.

---

> > > ### Comment · Reviewer_mLpm · 2024-12-03
> > >
> > > Dear authors,
> > >
> > > I appreciate the efforts and response. I was not looking for the results that demonstrate the effectiveness of the proposed method under different scenarios that I mentioned. I wanted to see analysis results and discussions as to why the proposed method works or does not work.
> > >
> > > One would expect the proposed method to perform poorly if ID samples blurry since the OOD samples are synthesized via blur. The authors have claimed that the proposed method works under such scenario (although I cannot completely agree with the experimental settings). I believe there should be detailed discussions as to why the proposed method works when it’s not expected to.

---

### Official Review · Reviewer_Y5ac · 2024-11-02

**Soundness:** 3
**Presentation:** 2
**Contribution:** 2
**Rating:** 5
**Confidence:** 3

**Summary:**

This paper aim to alleviate a well-known weakness of normalizing flows models towards detecting complexity for the task of out-of-distribution (OOD) detection. To accomplish this, the presented methodology is composed of three changes. First, the method applies gaussian blur to the original in-distribution (ID) images to generate synthetic OOD images. Second, the method add an additional softplus function to the maximum likelihood objective to prevent numerical instability. Lastly, the method construct a new OOD score that is the combination of the predicted likelihood and the complexity of the image. The paper reveals that these modification improved the OOD detection capabilities of the normalizing flow model considerably.

**Strengths:**

1. The motivation and methodology design is concise and very written.
2. The dataset used for evaluation is comprehensive: SVNH, LSUN, CelebA, CIFAR-10, CIFAR-100 for visual recognition; Chest X-ray, RealBlur, and KonIQ-10K for high-resolution imaging, and movie reviews, AG News, SST-2, and WikiText-2 for text.
3. Detection performance shows improvement from the addition of synthetic outliers and complexity scoring.

**Weaknesses:**

1. The in-distribution performance is completely missing in the evaluation. While the paper shows that adding synthetic OOD images via blurring improves detection performance, how does it affect the normalizing flow's ability to model the ID data distribution (i.e. FID between the generated images and ID images)?
2. The performance gains from using synthetic outliers is very marginal compared to complexity score. It is unclear whether it is even necessary (especially considering 1.) because after adding the complexity score (shown in Table 3), the effect of adding synthetic data is very marginal (sometime even slightly worse than MLE in the case of SVNH).
3. The significance and novelty with the addition of the complexity score is weak since it is underexplored in this work. The usage of JPEG2000 compression to calculate complexity limits its uses to only image data, and hence, the lack of evaluation of complexity scores on high-resolution image data and text data.
4. The paper does not provide literature support of the claim "synthetic outliers enhance the local Lipschitz constant, improving model stability and performance" in line 432-437. This claim is counterintuitive: the Lipschitz constant provides an upper bound on the change in model prediction, which is looser with higher constant. i.e. prediction can be more sensitive towards small perturbations, and hence, less stable. Empirical evidence or theoretical justification is needed to justify this counterintuitive claim.

**Questions:**

1. The paper claim to estimate Lipschitz Constant by taking the L_infinity norm of the gradient vector. How are the samples selected here?
2. What is the slope and R^2/correlation of likelihood vs. complexity in Figure 3? It seems like the relationship between likelihood and complexity is rather weak.

---

> ### Author Response · Authors · 2024-11-26
> **Response to Reviewer Y5ac (1/2)**
>
> Thank you for your detailed and thoughtful review. We are grateful for your suggestions, which have helped us improve the clarity and rigor of our work. Below, we address your specific points:
>
> **Weakness 1**: The in-distribution performance is completely missing in the evaluation. While the paper shows that adding synthetic OOD images via blurring improves detection performance, how does it affect the normalizing flow's ability to model the ID data distribution (i.e., FID between the generated images and ID images)?
>
> **Response**: We have added an evaluation of in-distribution performance (Figure 4) by examining the likelihood scores of in-distribution data, as this aligns with the task-oriented nature of our approach. The experimental results indicate that the likelihood scores of in-distribution data are not adversely affected by the inclusion of synthetic out-of-distribution points. This demonstrates that our method improves OOD detection performance while maintaining the capability of the in-distribution data modeling.
>
> **Weakness 2**: The performance gains from using synthetic outliers are very marginal compared to the complexity score. It is unclear whether it is even necessary because after adding the complexity score (shown in Table 3), the effect of adding synthetic data is very marginal (sometimes even slightly worse than MLE in the case of SVNH).
>
> **Response**: For complexity-based methods to be effective, the complexity of OOD data must be lower than that of ID data. Therefore, relying solely on complexity can lead to instability, as seen in the poor results when CIFAR-10 is used as the ID dataset and iSUN as the OOD dataset. Our method provides a more stable baseline under such conditions. Furthermore, we add new experiemtns by  incorporating outlier augmentation techniques (e.g., CutPaste, CutMix, Mixup) significantly improves performance, highlighting the robustness and effectiveness of our approach using synthetic outliers.
>
> **Weakness 3**: The significance and novelty with the addition of the complexity score is weak since it is underexplored in this work. The usage of JPEG2000 compression to calculate complexity limits its uses to only image data, and hence, the lack of evaluation of complexity scores on high-resolution image data and text data.
>
> **Response**: In the paper, we did present the results with high-resolution images, such as  Chest X-ray and MVTec dataset. JPEG2000's scalability allows for effective complexity analysis across varied image resolutions. Regarding text data, as outlined in our paper, we utilize lossless compression algorithms such as gzip to evaluate complexity of the entire dataset. This approach remains consistent with our framework.
>
> **Weakness 4**: The paper does not provide literature support for the claim "synthetic outliers enhance the local Lipschitz constant, improving model stability and performance" in line 432-437. This claim is counterintuitive: the Lipschitz constant provides an upper bound on the change in model prediction, which is looser with a higher constant. i.e., prediction can be more sensitive to small perturbations, and hence, less stable. Empirical evidence or theoretical justification is needed to justify this counterintuitive claim.
>
> **Response**: Our claim regarding the enhancement of the local Lipschitz constant $L_{\mathcal{A}}$ through the use of synthetic outliers is primarily based on the Hypothesis 1 in page 3 that correlates the Lipschitz constant with image complexity. Specifically, the hypothesis  suggests that by increasing $L_{\mathcal{A}}$, when the complexity $C(\mathbf{x})$of the input decreases, the model tolerate a large range of  $\mathbb{P}\left(\mathcal{B}_{\mathbf{z}}^\epsilon\right)$.
>
> However, we acknowledge that this hypothesis provides a theoretical perspective that lacks rigorous proof, and therefore, we cannot claim definitive theoretical support. To substantiate our claim, we conducted empirical experiments focusing on the performance metrics, particularly the AUROC scores. These experiments demonstrated that introducing synthetic outliers not only improved the AUROC scores but also resulted in an increased $L_{\mathcal{A}}$. These findings, while initially based on theoretical conjectures, are thus empirically validated to some extent, supporting the effectiveness of our methodology in practice.

---

> ### Author Response · Authors · 2024-11-26
> **Response to Reviewer Y5ac (2/2)**
>
> **Question 1** : The paper claim to estimate Lipschitz Constant by taking the L_infinity norm of the gradient vector. How are the samples selected here?
>
> **Response**: We randomly select 1,000 samples from a normal distribution representing the input space. Each sample $\mathbf{x}_i$ is then used to compute the gradient of the model’s output $f(\mathbf{x}_i)$ with respect to the input, utilizing PyTorch’s automatic differentiation tool. The L2 norm $\left\||\nabla f\left(\mathbf{x}_i\right)\right\||$ of each gradient is calculated. We then approximate the Lipschitz constant by taking the maximum gradient norm observed across all samples, $\max_i \left\||\nabla f\left(\mathbf{x}_i\right)\right\||$.
>
>
> **Question2** : What is the slope and R^2/correlation of likelihood vs. complexity in Figure 3? It seems like the relationship between likelihood and complexity is rather weak.
>
> **Response**: In Figure 3, the relationship between model-predicted likelihood and image complexity is depicted for ID data (blue points) and OOD data (green and red points) on a scatter plot, with complexity on the x-axis and likelihood on the y-axis. The pink line represents the linear regression fit to these data points. This figure was created to highlight how the model corrects the bias between predicted likelihood and image complexity, underscoring the effectiveness of our approach. In the CIFAR-10 dataset, although the linear trend between complexity and likelihood is not strongly modified, we emphasis that there is a clear decline in likelihood for OOD data points. Moreover, for the CIFAR-100 dataset, the likelihood values do not exhibit a significant bias with respect to complexity, showing that the model effectively maintains stability across varying levels of image complexity. These observations validate our method's effectiveness in managing complexity biases across different datasets.
>
> Your comprehensive feedback has greatly helped us address key gaps and improve the quality of the paper. Thank you again for your valuable review.

---

> > ### Comment · Reviewer_Y5ac · 2024-11-27
> >
> > Thank you for the response. I have read the author's response to my and other reviewers' comments. I believe most of my major concerns were addressed, and therefore, I will be raising my score to a 5. I do believe the presentation of the paper can be further improved to avoid some of my confusions: i.e. connecting Lipschitz constant back to Hypothesis 1 since one is on page 9 and the other on page 3, present the inclusion/exclusion of complexity scoring as an ablation rather the main result in Table 2. Also minor suggestion is presentation of newly added Figure 4: since x-axis isn't meaningful, a KDE plot could make the point clearer.

---

> > > ### Author Response · Authors · 2024-11-28
> > >
> > > Thank you for your feedback. Regarding the points you raised:
> > >
> > > 1. Hypothesis 1 primarily motivated our design of low-complexity outliers, and the experiments serve as a validation of this motivation. We will further clarify the connection between Hypothesis 1 and the experiments.
> > > 2. Thank you for your suggestion regarding the presentation of complexity scoring. Due to the limited time available, we will make this change in a future version of the manuscript.
> > > 3. We have uploaded the new version of the manuscript, which includes the revised Figure 4 with a KDE plot. The KDE plot provides a more intuitive visual representation of the ID likelihood.
> > >
> > > We truly appreciate your constructive feedback and the time you took to help improve the paper.

---

### Official Review · Reviewer_kh3v · 2024-11-03

**Soundness:** 2
**Presentation:** 2
**Contribution:** 2
**Rating:** 5
**Confidence:** 4

**Summary:**

This paper studies the problem of OOD detection, specifically focusing on correcting the likelihood bias in normalizing flows that affects their performance in OOD detection. The authors propose incorporating synthetic outliers during training and introduce an adversarial likelihood objective, utilizing the softplus function to improve model stability. Experiments on both benchmark datasets and high-dimensional real-world datasets show that the proposed method achieves significant improvements in OOD detection accuracy, yielding results comparable to models trained with limited real outlier data.

**Strengths:**

- The paper proposes a novel approach by using synthetic outliers to correct bias in normalizing flows for OOD detection, uniquely addressing data complexity issues. The softplus-based objective further enhances model stability, setting this method apart in improving robustness.

- The experimental setup is thorough, including benchmarks across various datasets and both image and text modalities. The results consistently demonstrate the effectiveness of the proposed approach, particularly in improving AUROC and other detection metrics.

- The paper is well-organized, and the methodology is explained in detail, making it easy for readers to understand the approach. The visualizations, such as complexity distributions and comparisons between standard and softplus-based training, add clarity to the results and methodology.

**Weaknesses:**

- The choice of synthetic outliers, particularly using Gaussian blur for images, may limit applicability to cases where blurring captures outlier characteristics. For complex datasets with nuanced OOD structures, more sophisticated synthetic outlier generation techniques may be needed.

- While the softplus objective enhances stability, it introduces additional computational overhead, especially in high-dimensional data scenarios. The paper could explore potential optimizations for large-scale datasets.

- The method’s performance is heavily tied to the complexity assumptions underlying the model. The complexity-adjusted scoring approach, while effective, might misinterpret data with atypical structures. Additional investigation into adapting complexity measures could broaden the method's applicability.

**Questions:**

- How does the method handle OOD data that exhibit similar complexity levels to the ID data but differ in semantic content? Could the current approach potentially overlook such cases?

- Is the choice of Gaussian blur for synthetic outlier generation extendable to other domains, or would you recommend domain-specific modifications for applications outside of vision and text?

- Could you elaborate on any observed trade-offs between using synthetic outliers versus real outliers in terms of computational efficiency and detection accuracy?

---

> ### Author Response · Authors · 2024-11-26
> **Response to Reviewer kh3v (1/2)**
>
> Thank you for your constructive and detailed feedback, which provided valuable insights into improving the applicability and robustness of our method. Your comments encouraged us to refine our synthetic outlier generation strategy and address complexity-related concerns. Below, we provide responses to your specific points:
>
> **Weakness 1**: The choice of synthetic outliers, particularly using Gaussian blur for images, may limit applicability to cases where blurring captures outlier characteristics. For complex datasets with nuanced OOD structures, more sophisticated synthetic outlier generation techniques may be needed.
>
> **Response**: We emphasis that Gaussian blurred samples are introduced for bias correction occured for normalized flow based OOD detection approaches and can be combined with other types of synthetic outliers to improve detection ablibity. In our new revised manuscript, we incorporated additional synthetic outlier generation techniques, including CutPaste, CutMix, and MixUp with Gaussian blur, which capture more diverse and nuanced OOD characteristics. These methods were validated on the benchmark datasets on base normalizing flow model and MVTec dataset using FastFlow and CS-Flow, demonstrating consistent improvements and better coverage of complex OOD scenarios.
>
> **Weakness 2**: While the softplus objective enhances stability, it introduces additional computational overhead, especially in high-dimensional data scenarios. The paper could explore potential optimizations for large-scale datasets.
>
> **Response**: We appreciate the reviewer's concern regarding the potential computational overhead introduced by the softplus objective, especially in high-dimensional data scenarios. However, we would like to clarify that the softplus transformation $\text{Softplus}(\log p_\mathcal{X}(\mathbf{x}^\prime))$ is applied only to the log likelihood $\log p_\mathcal{X}(\mathbf{x}^\prime)$ (which is a scalar), not to the full high-dimensional data or intermediate representations. Therefore, the computational cost introduced by this operation is negligible, even in large-scale datasets or high-dimensional settings.
>
> **Weakness 3**: The method’s performance is heavily tied to the complexity assumptions underlying the model. The complexity-adjusted scoring approach, while effective, might misinterpret data with atypical structures. Additional investigation into adapting complexity measures could broaden the method's applicability.
>
> **Response**: It's important to clarify that for the training phase, our model does not contain complexity scores as the input. For the testing phase, if the complexity of the test data significantly exceeds that seen in training, we advise relying on likelihood alone for out-of-distribution detection. This adjustment is suggested to mitigate potential misinterpretations caused by complexity measures, thereby improving the method’s accuracy and applicability.
>
> **Question 1**: How does the method handle OOD data that exhibit similar complexity levels to the ID data but differ in semantic content? Could the current approach potentially overlook such cases?
>
> **Response**: Our method is designed not merely to utilize complexity measures for detecting OOD data but primarily to correct the complexity bias inherent in normalizing flows. The experimetns demonstrates that  such data augemntation helps distinguish between ID and OOD data more effectively. For OOD data with similar complexity to ID data but differing in semantic content, as seen with CIFAR-10/CIFAR-100 and LSUN, our synthetic outlier augmentation significantly improves performance. It boosts AUROC from 86.6 to 96.0 for CIFAR-10 and from 67.5 to 89.1 for CIFAR-100, demonstrating its effectiveness.
>
> **Question 2**: Is the choice of Gaussian blur for synthetic outlier generation extendable to other domains, or would you recommend domain-specific modifications for applications outside of vision and text?
>
> **Response**: For other domains, such as signal Gaussian blur can also be adapted. For example, smoothing a signal through low-pass filtering serves a similar purpose by reducing high-frequency details and overall complexity. While in the text domain, we use synonym substitution to alter semantic content without dramatically changing structure, and we simplify complexity by removing subordinate clauses or reducing sentence length. These transformations provide meaningful shifts in textual complexity while preserving domain relevance.
>
> In summary, while Gaussian blur itself may not always be directly applicable, the overarching principle of controlled complexity modification is extendable across domains. With appropriate domain-specific transformations, this approach can effectively generate meaningful synthetic outliers for a wide variety of data types.

---

> ### Author Response · Authors · 2024-11-26
> **Response to Reviewer kh3v (2/2)**
>
> **Question 3**: Could you elaborate on any observed trade-offs between using synthetic outliers versus real outliers in terms of computational efficiency and detection accuracy?
>
> **Response**: In our experiments, we separately employ either real or synthetic outliers to compare their effectiveness in detecting OOD data. Thus we do not have the problems of trade-off between real and syntheis ones. Our experimetents show that under conditions that real outliers are not avaible, our method with synthetic ones achieve the similar performance as the model trainned with real outliers.

---

### Meta-Review · Area_Chair_RL7b · 2024-12-20

**Metareview:**

The paper focuses on likelihood-based OOD detection in normalizing flows. The authors propose to train normalizing flows on a mix of positive likelihood on IID samples and negative softplus of likelihood on synthetic outliers. Synthetic outliers are generated by combining data augmentation and Gaussian blur for images, and an analogous technique for language task. The method improves OOD detection performance across a broad range of evaluations.

Strengths:
- The proposed method is very simple and makes intuitive sense.
- The experiments in the paper are very extensive. Multiple new experiments have been added during the rebuttal phase to address reviewer concerns.
- The paper provides additional insight into the complexity bias of generative models.

Weaknesses:
- The novelty of the paper is fairly limited. Prior work has shown that normalizing flows (as well as other density estimation models) are often assigning higher likelihood to out-of-distribution data compared to in-distribution data [1]. Prior work also showed that input complexity is highly correlated with the likelihood according to generative models and can be used to improve OOD detection [2]. [3] proposed outlier exposure as a way of improving OOD detection. [4], in particular, showed results on outlier exposure with normalizing flows. The core contribution is in the combination of (1) specific pipeline for generating outliers, (2) combining outlier training with the complexity penalty at detection stage, and (3) broad evaluation.
- The outlier generation pipeline works for the benchmark datasets, but would likely fail if the data distribution is invariant to bluring. I.e. if the dataset contains both images $x$ and their blured versions, for various bluring strengths.
- While the method provides consistent improvements, it does not necessarily advance state of the art for OOD detection.

Decision recommendation: The reviewers were unanimous in recommending a weak reject. I believe the paper shows interesting practical results, and the proposed method is simple and fairly practical. However, the novelty of the observations is fairly limited, and the method is likely to fail on some types of data distributions (with broad ranges of complexity, invariant to bluring, etc). Thus, I am leaning towards rejecting the paper.

[1] Do Deep Generative Models Know What They Don't Know?
Eric Nalisnick, Akihiro Matsukawa, Yee Whye Teh, Dilan Gorur, Balaji Lakshminarayanan

[2] Input complexity and out-of-distribution detection with likelihood-based generative models
Joan Serrà, David Álvarez, Vicenç Gómez, Olga Slizovskaia, José F. Núñez, Jordi Luque

[3] Deep Anomaly Detection with Outlier Exposure
Dan Hendrycks, Mantas Mazeika, Thomas Dietterich

[4] Why Normalizing Flows Fail to Detect Out-of-Distribution Data
Polina Kirichenko, Pavel Izmailov, Andrew Gordon Wilson

**Additional Comments On Reviewer Discussion:**

All four reviewers gave a score of 5 to the paper. The authors provided multiple new results during the rebuttal phase, but none of the reviewers were convinced by these results to accept the paper. Some of the concerns included limited originality, the relative importance of the complexity score, the fact that the method shouldn't help if the data distribution includes images with various levels of blur.

---

### Decision · Program_Chairs · 2025-01-22

Reject